# PRIOR KNOWLEDGE REPRESENTATION FOR SELF-ATTENTION NETWORKS

## ABSTRACT

Self-attention networks (SANs) have shown promising empirical results in various natural language processing tasks. Typically, it gradually learning language knowledge on the whole training dataset in parallel and stacked ways, thereby modeling language representation. In this paper, we propose a simple and general representation method to consider prior knowledge related to language representation from the beginning of training. Also, the proposed method allows SANs to leverage prior knowledge in a universal way compatible with neural networks. Furthermore, we apply it to one prior word frequency knowledge for the monolingual data and other prior translation lexicon knowledge for the bilingual data, respectively, thereby enhancing the language representation. Experimental results on WMT14 English-to-German and WMT17 Chinese-to-English translation tasks demonstrate the effectiveness and universality of the proposed method over a strong Transformer-based baseline.

## 1 INTRODUCTION

Self-attention networks (SANs) have attracted increasing attention in the natural language processing community. Instead of using complex recurrent or convolutional neural networks (Sutskever et al., 2014; Bahdanau et al., 2015), SANs first use the positional encoding mechanism Gehring et al. (2017) to encode order dependencies in the language. The learned positional embedding is then added to corresponding word embedding to obtain an input representation, based on which SANs perform (multi-head) and stack (multi-layer) self-attentive functions (Vaswani et al., 2017) in parallel to learn language representation. The SAN-based models are iteratively optimized to model language knowledge on the whole training dataset, which has achieved state-of-the-art performance in many natural language processing tasks pairs (Barrault et al., 2019; Oepen et al., 2019; Weissenbacher & Gonzalez-Hernandez, 2019; Demner-Fushman et al., 2019).

Despite the success, SANs gradually model language knowledge on the batch-level datasets and does not consider the prior knowledge on the whole dataset from the beginning of training, which may decrease its language representation capability. For example, the SAN-based neural machine translation (NMT) model often mistranslate into words that seem to be natural in the target language sentence, but do not reflect the original meaning of the source language sentence (Arthur et al., 2016; Wang et al., 2017a). As a result, the NMT model produces fluent yet sometimes inadequate translations (Tu et al., 2016; 2017). To address this issue, recent studies explored the prior knowledge which has the stringer ability to model the fluency of translation in traditional SMT (Koehn et al., 2003; Liu et al., 2006; Chiang, 2007; Liu et al., 2007). Take the prior translation lexicon knowledge as an example, Arthur et al. (2016) directly biased or interpolated the bilingual lexicon translation distribution with the output of the softmax layer of NMT to improve the translations of infrequency words. An auxiliary classifier (Zhao et al., 2018; Wang et al., 2018) were employed to integrate the SMT recommendations with NMT generations to generate the faithful translation. In addition, the phrase translation rules was as the recommendation memory to make better predictions in NMT (Wang et al., 2017b; Zhao et al., 2018). Although these studies successfully improved the issue of inadequate translations in NMT, they tended to focus on exploring the prior translation lexicon knowledge by using their specific methods. In other words, these unique methods make it difficult to explore other prior knowledge in a universal way and to determine which of the prior knowledge and the unique method this improvement comes from. Meanwhile, these studies directly

utilized the probability distribution of the prior knowledge and lacked the neural network's ability to semantically generalize, while will further hinder the language representation ability of SANs.

In this paper, we propose a simple and general representation method to introduce the prior knowledge into SANs. In particular, we package the prior knowledge related with one source sentence to a continuous space matrix, which allows SANs to utilize the prior knowledge from the beginning of training, thereby better performing language representation in a universal way compatible with neural networks. To maintain the simplicity and flexibility of the SANs, we use the prior knowledge representation in parallel and stacked ways to learn the representation of the input sentence. Furthermore, we use the proposed method to explore one prior word frequency knowledge for the monolingual data and other prior translation lexicon knowledge for the bilingual data, respectively. Empirical results on two widely used translation data sets, including WMT14 English→German and WMT17 Chinese→English, to verify the effectiveness and universality of the proposed method over a strong Transformer-based baseline.

## 2 SELF-ATTENTION NETWORKS

The self-attention networks (SANs) (Vaswani et al., 2017) is composed of a stack of $N$ identical layers, each of which includes two sub-layers. Formally, given a input sentence with the length $J$, $X=\{x_1, x_2, \cdots, x_J\}$, the positional encoding mechanism (Gehring et al., 2017) is used to compute a positional embedding of each word based on its position index. The positional embedding is then added to the corresponding word embedding as an combined embedding, thereby gaining a sequence of input representation $\mathbf{H}^0=\{\mathbf{v}_1, \mathbf{v}_2, \cdots, \mathbf{v}_J\}$. Moreover, the stacked SANs is organized as follows:

$$
\begin{aligned}
\overline{\mathbf{H}}^n &= \mathrm{LN}(\mathrm{SelfAtt}^n(\mathbf{Q}^{n-1}, \mathbf{K}^{n-1}, \mathbf{V}^{n-1}) + \mathbf{H}^{n-1}), \\
\mathbf{H}^n &= \mathrm{LN}(\mathrm{FFN}^n(\overline{\mathbf{H}}^n) + \overline{\mathbf{H}}^n),
\end{aligned}
\tag{1}
$$

where $\mathrm{SelfAtt}^n(\cdot)$, $\mathrm{LN}(\cdot)$, and $\mathrm{FFN}^n(\cdot)$ are self-attention module, layer normalization (Ba et al., 2016), and feed-forward network for the $n$-th identical layer, respectively. $\mathbf{Q}^{n-1}$, $\mathbf{K}^{n-1}$, and $\mathbf{V}^{n-1}$ are query, key, and value matrices that are transformed from the ($n$-1)-th layer $\mathbf{H}^{n-1}$. For example, $\mathbf{Q}^0$, $\mathbf{K}^0$, and $\mathbf{V}^0$ are packed from the $\mathbf{H}^0$ learned by the positional encoding mechanism (Gehring et al., 2017). In particular, $\mathrm{SelfAtt}_n(\cdot)$ is applied on the $\{\mathbf{Q}^{n-1}, \mathbf{K}^{n-1}, \mathbf{V}^{n-1}\}$ of the $n$-1 layer:

$$
\mathrm{SelfAtt}^n(\mathbf{Q}^{n-1}, \mathbf{K}^{n-1}, \mathbf{V}^{n-1}) = \mathrm{softmax}(\mathbf{Q}^{n-1}\mathbf{K}^{n-1\top}/\sqrt{d_{model}})\mathbf{V}^{n-1},
\tag{2}
$$

where $d_{model}$ is the dimension size of the query and key vectors. As a result, the output of the $N$-th layer $\mathbf{H}^N$ is the representation of the input sentence. Moreover, the self-attention mechanism can be further refined as multi-head self-attention to jointly attend to the information from different representation sub-spaces at different positions.

## 3 PRIOR KNOWLEDGE REPRESENTATION

In this section, we propose a simple and general representation method to encode the prior knowledge, which allows SANs to model the prior knowledge in a manner compatible with neural networks. Given a input sentence $X=\{x_1, x_2, \cdots, x_J\}$ with the length $J$, we represent the associated prior knowledge as a matrix $\mathbf{M}$:

$$
\mathbf{M} = \begin{bmatrix}
\mathbf{m}_1^1 & \mathbf{m}_1^2 & \cdots & \mathbf{m}_1^K \\
\mathbf{m}_2^1 & \mathbf{m}_2^2 & \cdots & \mathbf{m}_2^K \\
\vdots & \vdots & \ddots & \vdots \\
\mathbf{m}_J^1 & \mathbf{m}_J^2 & \cdots & \mathbf{m}_J^K
\end{bmatrix},
\tag{3}
$$

where each row denotes the prior knowledge related with word $x_j$ and each element $\mathbf{m}_j^t$ is a fixed size vector. Also, $\mathbf{M}$ is packed into a key and value matrix pair $\{\mathbb{K}, \mathbb{V}\}$ for the prior knowledge. The prior $\{\mathbb{K}, \mathbb{V}\}$ and the current $\mathbf{Q}$ are the input to the self-attention mechanism (see Eq.(2)) to learn a prior knowledge representation $\mathbf{PK}$ for the input sentence $X$:

$$
\mathbf{PK} = \mathrm{LN}(\mathrm{SelfAtt}(\mathbf{Q}, \mathbb{K}, \mathbb{V}) + \mathbf{H}),
\tag{4}
$$

where $\mathbf{Q}$ is transformed from the current sentence representation $\mathbf{H}$, for example, the $n$-th layer output $\mathbf{H}^n$ of the stacked SANs in Eq.(1). Finally, $\mathbf{PK}$ is the expected prior knowledge representation (PKR). Later, we will apply the proposed method to one prior word frequency knowledge for the monolingual data and other prior translation lexicon knowledge for the bilingual data, respectively, thereby enhancing the language representation.

## 3.1 WORD FREQUENCY BASED PRIOR KNOWLEDGE REPRESENTATION

Recent studies have shown the effectiveness of the prior word frequency information for improving the translations of content words (Arthur et al., 2016; Wang et al., 2018; He et al., 2019; Chen et al., 2020). Intuitively, these content words generally have more meanings and lower word frequency than function words in one sentence. To capture the prior knowledge, we use word frequency information to enhance the importance of content words in the sentence representation. Inspired by the previous studies (Setiawan et al., 2007; 2009; Zhang & Zhao, 2013; Chen et al., 2020), we use the frequency of each word $w$ on the monolingual corpus $F$ to distinguish content words and function words in the input sentence. To this end, let the $B$ most frequent words in the monolingual corpus denote the function words while the remaining words are regarded as content words for the input sentence. In other words, let the prior word frequency knowledge of each word denote a binary mask:

$$f_j = \begin{cases} 0, & x_j \in B \\ 1, & x_j \notin B \end{cases} \tag{5}$$

Furthermore, the prior word frequency knowledge of the input sentence is a diagonal matrix according to Eq.(3):

$$\mathbf{M}_F = \begin{bmatrix} f_1 & 0 & \cdots & 0 \\ 0 & f_2 & \cdots & 0 \\ \vdots & \vdots & \ddots & \vdots \\ 0 & 0 & \cdots & f_J \end{bmatrix}. \tag{6}$$

Finally, we use $\mathbf{M}_F$ to mask the representation of function words in $\mathbf{H}$, and thereby gain key and value matrices $\{\mathbb{K}_F, \mathbb{V}_F\}$ to learn a prior word frequency based PKR $\mathbf{PK}_F$ for the input sentence:

$$\mathbf{PK}_F = \mathrm{LN}(\mathrm{SelfAtt}(\mathbf{Q}, \mathbb{K}_F, \mathbb{V}_F) + \mathbf{H}), \tag{7}$$

where $\mathbf{Q}$ is transformed from the current sentence representation $\mathbf{H}$, for example, $\mathbf{H}^n$ in Eq.(1).

## 3.2 BILINGUAL TRANSLATION LEXICON BASED PRIOR KNOWLEDGE REPRESENTATION

We apply the proposed PKR method into the prior bilingual translation lexicons in SMT (Brown et al., 1993). Discretely, each source word $x_j$ has multiple associated target translation options retrieved from the bilingual translation lexicon table. For simplicity, we select the top $L$ target translation options as the prior translation knowledge of word $x_j$ are remained according to their lexicon translation probabilities and the entry corresponds to the target word. Formally, all retrieved lists for the input sentence $X$ are constructed as a $J \times K \times d_{model}$ prior translation knowledge matrix $\mathbf{M}_T$ according to Eq.(3):

$$\mathbf{M}_T = \begin{bmatrix} \mathbf{t}_1^1 & \mathbf{t}_1^2 & \cdots & \mathbf{t}_1^L \\ \mathbf{t}_2^1 & \mathbf{t}_2^2 & \cdots & \mathbf{t}_2^L \\ \vdots & \vdots & \ddots & \vdots \\ \mathbf{t}_J^1 & \mathbf{t}_J^2 & \cdots & \mathbf{t}_J^L \end{bmatrix}. \tag{8}$$

where each row corresponds to the retrieved list of the word $x_j$, and the entry corresponds to the embedding of the target word. In particular, when the number of the associated target words for one source word $x_j$ is less than $L$, we use the embedding of a special placeholder "/" to fill the $j$-th row in $\mathbf{M}_T$ up to $L$. Also, the $\mathbf{M}_T$ is packed into a key and value matrix pair $\{\mathbb{K}_T, \mathbb{V}_T\}$. $\{\mathbf{Q}, \mathbb{K}_T$, and $\mathbb{V}_T\}$ are the input to Eq.(4) to learn a prior translation knowledge representation $\mathbf{PK}_T$ for the source sentence:

$$\mathbf{PK}_T = \mathrm{LN}(\mathrm{SelfAtt}(\mathbf{Q}, \mathbb{K}_T, \mathbb{V}_T) + \mathbf{H}), \tag{9}$$

where $\mathbf{Q}$ is transformed from the current sentence representation $\mathbf{H}$, for example, $\mathbf{H}^n$ in Eq.(1).

## 4 SANS WITH PRIOR KNOWLEDGE REPRESENTATION

Intuitively, the prior knowledge assists SANs to enhance the language representation in the SAN-based models of NLP tasks. In other words, the prior knowledge plays an auxiliary role during the processing of learning language representation. Therefore, we compute a gate scalar $g^n \in [0,1]$ to weight the expected importance of the prior knowledge representation **PK** at the $n$-th layer:

$$g^n = \mathrm{Sigmoid}(\mathbf{U}_g^n \mathbf{H}^n + \mathbf{W}_g^n \mathbf{PK}), \quad (10)$$

where the Sigmoid is an active function, and $\mathbf{W}_g^n \in \mathbb{R}^{d_{model} \times 1}$ and $\mathbf{U}_g^n \in \mathbb{R}^{d_{model} \times 1}$ are model parameters. We then fuse $\mathbf{H}^n$ and **PK** through the gate scalar $g^n$ to learn the sentence representation at the $n$-th layer:

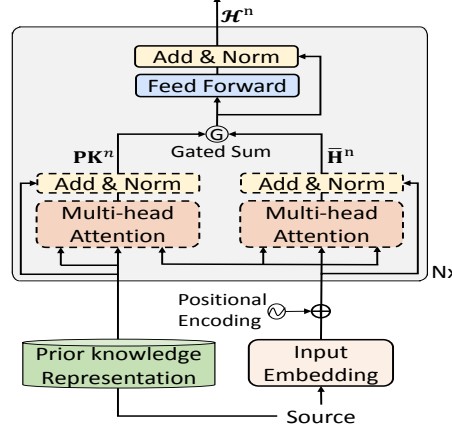

Figure 1: SANs with prior knowledge representation. Dotted boxes denote the shared modules.

$$\boldsymbol{\mathcal{H}}^n = \mathbf{H}^n + g^n \cdot \mathbf{PK}. \quad (11)$$

Also, the new stacked SAN is formally denoted as the modified version of Eq.(1):

$$\overline{\mathbf{H}}^n = \mathrm{LN}(\mathrm{SelfAtt}^n(\mathbf{Q}^{n-1}, \mathbf{K}^{n-1}, \mathbf{V}^{n-1}) + \boldsymbol{\mathcal{H}}^{n-1}),$$
$$\mathbf{PK}^n = \mathrm{LN}(\mathrm{SelfAtt}^n(\mathbf{Q}^n, \mathbb{K}, \mathbb{V}) + \overline{\mathbf{H}}^n),$$
$$g^n = \mathrm{Sigmoid}(\mathbf{U}_g^n \overline{\mathbf{H}}^n + \mathbf{W}_g^n \mathbf{PK}^n), \quad (12)$$
$$\overline{\boldsymbol{\mathcal{H}}}^n = \overline{\mathbf{H}}^n + g^n \cdot \mathbf{PK}^n,$$
$$\boldsymbol{\mathcal{H}}^n = \mathrm{LN}(\mathrm{FFN}^n(\overline{\boldsymbol{\mathcal{H}}}^n) + \overline{\boldsymbol{\mathcal{H}}}^n),$$

where $\boldsymbol{\mathcal{H}}^0$ is the initialized sentence representation $\mathbf{H}^0$ as in the Section 2. As a result, there is a more effective sentence representation $\boldsymbol{\mathcal{H}}^N$ for the input sentence. Later, $\boldsymbol{\mathcal{H}}^N$ will be fed into the *Decoder* of the SAN-based model to model NLP tasks, for example, machine translation task.

## 5 EXPERIMENTS

### 5.1 EXPERIMENT SETUP

In this paper, we selected the machine translation task to evaluate the effectiveness of the proposed method, that is, two widely-used datasets: The WMT14 En-De includes 4.43 million bilingual sentence pairs, and the *newstest2013* and *newstest2014* datasets were used as the validation set and test set, respectively; The WMT17 Zh-En includes 22 million bilingual sentence pairs, and the *newsdev2017* and *newstest2017* datasets were used as the validation set and the test set, respectively. In the experiment, we first used the *fast_align* toolkit Dyer et al. (2013) to obtain word alignments from the source language to the target language. We then learned bilingual translation lexicon table between each word pair from the bilingual parallel training data with word alignments.

The byte pair encoding algorithm (Sennrich et al., 2016) was adopted, and the vocabulary size was set to 40K. For the Transformer base NMT, the dimension of all input and output layers was set to 512, the dimension of the inner feedforward neural network layer was set to 2048, and the total heads of all multi-head modules were set to 8 in both the encoder and decoder layers. In each training batch, there was a set of sentence pairs containing approximately $4096 \times 8$ source tokens and $4096 \times 8$ target tokens. During training, the value of label smoothing was set to 0.1, and the attention dropout and residual dropout were $p = 0.1$. The learning rate was varied under a warm-up strategy with warmup steps of 8,000. For evaluating the test sets, we used a beam size of 4 for decoding, and evaluated tokenized case-sensitive BLEU with the averaged model of the last 5 checkpoints for Transformer base model and 20 checkpoints for Transformer big model saved with an interval of 2,000 training steps. Following the training of 300,000 batches, the model with the highest BLEU score for the validation set was selected to evaluate the test sets. For the other setting not mentioned,

we followed the setting in Vaswani et al. (2017). All models were trained on eight V100 GPUs and evaluated on a single V100 GPU. The multi-bleu.perl script was used as the evaluation metric for the three translation tasks, and signtest (Collins et al., 2005) was used as the statistical significance test. We implemented the proposed NMT models on the *fairseq* toolkit (Ott et al., 2019).

## 5.2 BASELINE SYSTEMS

**Trans.base/big**: a vanilla Transformer-based base/big models (Vaswani et al., 2017). Both Trans.base and Trans.big models differ at the hidden size (512 vs. 1024), filter size (2048 vs. 4096), and the number of attention heads (8 vs. 16).

**+Shared-private**: It proposes shared-private bilingual word embeddings (Liu et al., 2019), which give a closer relationship between the source and target embeddings in the Transformer-based NMT.

**+SoftPrototype**: Each word in the input sequence is mapped into a distribution over the target vocabulary, and the weighted average of target word embeddings is treated as an "expected" word representation in the prototype R for the Transformer-based NMT model. It allows the *Decoder* to have indirect access to both past and future information (Wang et al., 2019).

**+D2GPo**: A kind of data-dependent Gaussian prior objective is proposed to build two probability distributions for the Transformer-based NMT, the first of which is from the detailed model training prediction and the second of which is from a ground-truth word-wise distribution (Li et al., 2020).

**+BCWAContLoss**: It used word frequency information to learn a sequence of content words for the source and target sentences. A content word-aware Transformer-based NMT model was designed to learn an additional content word-aware source representation and to utilize target content words to compute an additional loss during the training (Chen et al., 2020).

**+Soft Template**: It used extracted templates from tree structures as soft target templates to guide the translation procedure. We incorporate the prior template information into the encoder-decoder framework to jointly utilize the templates and source text to enhance the Transformer-based NMT (Yang et al., 2020).

## 5.3 TRANSLATION RESULTS

| Systems | En-De | | | Zh-En | |
|---|---|---|---|---|---|
| | BLEU | #Speed. | #Para. | BLEU | #Para. |
| *Existing NMT systems* | | | | | |
| Trans.base (Vaswani et al., 2017) | 27.3 | N/A | 65.0M | N/A | N/A |
| +Shared-private (Liu et al., 2019) | 28.06 | N/A | 65.0M | N/A | N/A |
| +D2GPo (Li et al., 2020) | 27.93 | N/A | N/A | N/A | N/A |
| +BCWAContLoss (Chen et al., 2020) | 28.51 | 13.1k | 72.8M | 24.94 | 81.0M |
| Trans.big (Vaswani et al., 2017) | 28.4 | N/A | 213.0M | N/A | N/A |
| +SoftPrototype (Wang et al., 2019) | 29.46 | N/A | 200.2M | N/A | N/A |
| +D2GPo (Li et al., 2020) | 29.10 | N/A | N/A | N/A | N/A |
| +BCWAContLoss (Chen et al., 2020) | 29.14 | 10.1k | 246.3M | 25.12 | 262.7M |
| +Soft Template (Yang et al., 2020) | 29.68 | N/A | N/A | N/A | N/A |
| *Our NMT systems* | | | | | |
| Trans.base | 27.67 | 13.2k | 66.5M | 24.28 | 74.7M |
| +$\mathbf{PK}_F$ | 28.41++ | 12.1k | 67.5M | 25.03++ | 75.7M |
| +$\mathbf{PK}_T$ | 28.66++ | 11.6k | 67.5M | 25.32++ | 75.7M |
| Trans.big | 28.65 | 11.2k | 221.2M | 24.84 | 237.5M |
| +$\mathbf{PK}_F$ | 29.21++ | 10.1k | 225.4M | 25.26 | 241.7M |
| +$\mathbf{PK}_T$ | 29.58++ | 9.1k | 225.4M | 25.54+ | 241.7M |

Table 1: Comparison of the proposed method with existing NMT systems on the three translation tasks. "#Speed." and "#Para." denote the training speed (tokens/second) and the size of model parameters, respectively. "+/++" after the score indicates that the proposed method was significantly better than the corresponding baseline Trans.base/big at significance level p<0.05/0.01.

**Main Results**: Table 1 showed the main results of WMT14 En-De and WMT17 Zh-En translation tasks. The BLEU scores of re-implemented Trans.base/big models were better than that of the original Trans.base/big models (Vaswani et al., 2017), which makes the evaluation convincing. As seen, our models (i.e., **+PK**$_F$ and **+PK**$_T$) and the comparison methods (+SoftPrototype, +Shared-private, +D2GPo, BCWAContLoss, and +Soft Template) were superior to the baseline Trans.base/big models. This indicates that introducing the prior knowledge consistently improved translation performance, demonstrating the effectiveness of the proposed prior knowledge representation approach.

**Evaluating Prior Knowledge**: Based on the proposed prior knowledge representation approach, the prior word frequency knowledge (**+PK**$_F$) and the prior translation lexicon knowledge (**+PK**$_T$) gained the improvement of BLEU scores on the Trans.base/big models. This indicates that the proposed approach provided a universal setting to better verify the effectiveness of two prior knowledge compare to the comparison models (i.e., +SoftPrototype, +Shared-private, +D2GPo, BCWAContLoss, and +Soft Template) with their specific methods. Furthermore, BLEU scores of **+PK**$_T$ were higher than that of **+PK**$_F$, that is, two types of prior knowledge have different improvements. The difference means that the improvement comes more from the prior knowledge itself rather than the method.

**Comparison with Previous Works**: Trans.base **+PK**$_F$ was superior to +Shared-privat and +D2GPo while it was slightly inferior to Trans.base +BCWAContLoss. **PK**$_F$ only focused on exploring the importance of source content words while BCWAContLoss captured the importance of source and target content words. In comparison, Trans.big **+PK**$_F$ was slightly superior to Trans.base +BCWAContLoss. The reason may be that the proposed approach can make full use of the prior word frequency knowledge on the Trans.big setting. **+PK**$_T$ outperformed +SoftPrototype, +Shared-private and +D2GPo and +BCWAContLoss on the Trans.base/big settings while it was slightly inferior to +Soft Template on the Trans.big seting. The reason may be that +Soft Template encoded the syntax constraint in addition to the prior translation knowledge.

**Model Parameters and Training Speed**: Take the En-De translation task as an example in Table 1, Trans.base **+PK**$_F$ and **+PK**$_T$ models contained approximately 1.5% additional parameters while both they decreased 8.3%/12.1% the training speeds, compared to the baseline Trans.base model. Meanwhile, Trans.base **+PK**$_T$ model achieved a comparable performance compared to the baseline Trans.big model which has many more parameters. This indicates that the improvement is indeed from prior knowledge rather than more parameters.

## 5.4 MODEL CONVERGENCE

In this section, we tried to verify whether the introduction of prior knowledge has an impact on the training process. Specifically, for Trans.base, **+PK**$_F$, and **+PK**$_T$ models, we observed their BLEU scores on the En-De valid set every five epoch intervals as shown in Figure 2. As seen, the learning curves of **+PK**$_F$ and **+PK**$_T$ are that of Trans.base, confirming the effectiveness of the prior translation lexicons and word frequency knowledge. Also, BLEU scores of **+PK**$_F$ and **+PK**$_T$ had been improving from the 5-th Epoch to the 55-th and 45-th Epochs, respectively. Thus, **+PK**$_F$ and **+PK**$_T$ reached the highest BLEU scores at the 55-th and 45-th Epochs. This means that **+PK**$_T$ converged faster than **+PK**$_F$ during

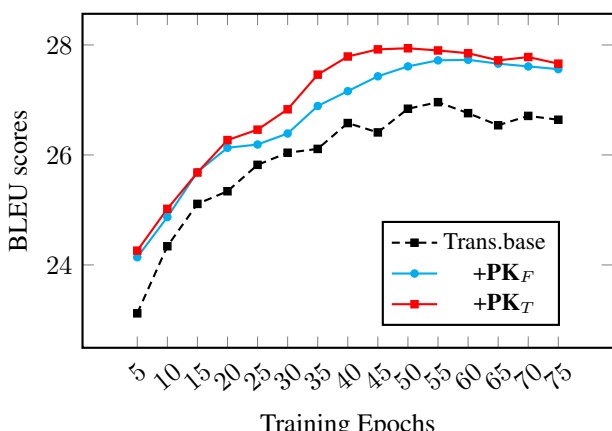

Figure 2: BLEU scores of different training epochs on the En-De valid set.

the training. This may be that the target language knowledge is advanced to the *Encoder* of NMT to learn a more effective representation of the source sentence.

## 5.5 PROBING EXPERIMENT

Following the word prediction experiment in (Weng et al., 2017)'s work, we also did a similar probing experiment here to evaluate the source representation learned by the proposed SANs with $\mathbf{PK}_F$ and $\mathbf{PK}_T$. In detail, for each sentence in the test set, we use the final source representation to make a prediction about the possible source words.

| System | Precision | | |
|---|---|---|---|
| | Top-200 | Top-500 | Top-1000 |
| Predictor of Trans.base | 45% | 55% | 66% |
| **+PK**$_F$ | 52% | 61% | 72% |
| **+PK**$_T$ | 61% | 72% | 79% |

Table 2: Precision of bag-of-words predictor.

possible source words. Given the set of top $C$ words in the target vocabulary and the set of words in all the references as $R$, and the precision of the word prediction: Precision $= |C \cap R|/|C| \times 100$. We then trained a *bag-of-words predictor* by maximizing $P(\mathbf{y}_{bow}|\mathcal{H}^N)$, where $\mathbf{y}_{bow}$ is an unordered set containing all target words in the output sentence. In addition to the structure of SANs with $\mathbf{PK}_F$ and $\mathbf{PK}_T$, the predictor included an additional feed-forward network layer which maps the final source sentence to the target word embedding matrix. Then, we compare the precision of target words in the top-$N$ words which are chosen through the predicted probabilities. As shown in Table 2, **+PK**$_F$ and **+PK**$_T$ gained higher precision in all conditions, especially **+PK**$_T$ was higher than **+PK**$_F$, which was consistent with the analysis that **+PK**$_T$ allows the target language knowledge to be advanced to the Encoder compare to **+PK**$_F$. This shows that the proposed method can obtain more information about the target language sentence and partial answers to why the proposed NMT models could improve generation quality.

## 5.6 EVALUATING HYPER-PARAMETERS IN $\mathbf{PK}_T$ AND $\mathbf{PK}_F$

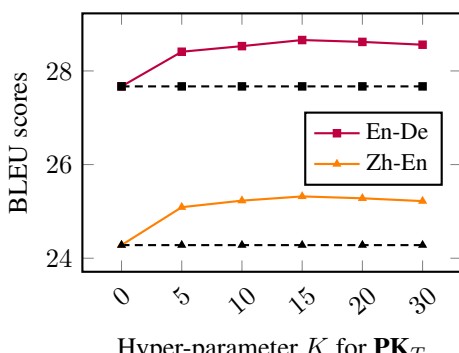
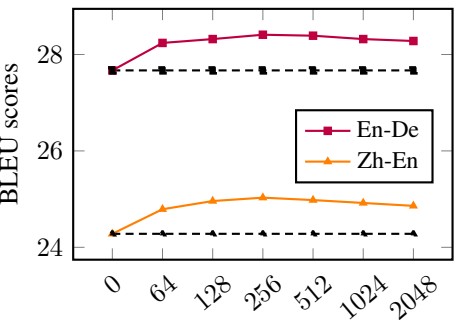

Figure 3: The left sub-figure shows BLEU scores of Trans.base+$\mathbf{PK}_T$ with the different hyper-parameter $K$; the right sub-figure shows BLEU scores of Trans.base+$\mathbf{PK}_F$ with the different hyper-parameter $B$. The dashed lines denote the results of the Trans.base model.

The left sub-figure of Figure 3 showed the results of Trans.base model (black dotted line) and **+PK**$_T$ model with different hyper-parameter $K$ for learning prior translation knowledge representation (colorful line) on the En-De and Zh-En test sets. When $K$ is one of (0, 5, 10, 15, 20, 30), the result of Trans.base+$\mathbf{PK}_T$ outperformed the Trans.base on the two test sets. Furthermore, Trans.base+$\mathbf{PK}_T$ reached the point of highest BLEU score in $K$=10, $K$=15, and $K$=10 on the En-De and Zh-En test sets, respectively. Finally, we trained the proposed NMT models in the two translation tasks (See Table 1) according to this optimized hyper-parameter $K$.

Furthermore, we masked the function words in the source sentence according to a list of the $B$ function words, thereby gaining the $\mathbf{M}_F$ in Eq.(6). The right sub-figure of Figure 3 showed the results of Trans.base+$\mathbf{PK}_F$ with the different number of the top $B$ function words on the En-De and Zh-En test sets. Trans.base+$\mathbf{PK}_F$ obtained the highest BLEU scores on both test sets over the Trans.base on modeling $B = 256$. The trained **+PK**$_F$ was as shown in Table 1.

### 5.7 TRANSLATING DIFFERENT LENGTH SENTENCES

To further evaluate our method, we showed the translation performance of source sentences with different sentence lengths. Specifically, we divided each test set into six groups according to the length of the source sentence, for example, "40" indicates that the length of sentences is between 30 and 40. Figure 4 shows the results of Trans.base, $+\mathbf{PK}_F$, and $+\mathbf{PK}_T$ models on the two translation tasks. As seen, $+\mathbf{PK}_F$ and $+\mathbf{PK}_T$ were superior to the baseline Trans.base in almost every length group on the two tasks, confirming the effectiveness of our method. Also, BLEU scores of all models decreased when the length was greater than thirty over the Zh-En task. BLEU scores of all models decreased when the length was greater than forty over the En-De task. The reason may be more diverse vocabulary in Chinese than English and German. Thus, a target word was mapped to more source words in the prior translation lexicons, that is, increasing the ambiguity of the target word.

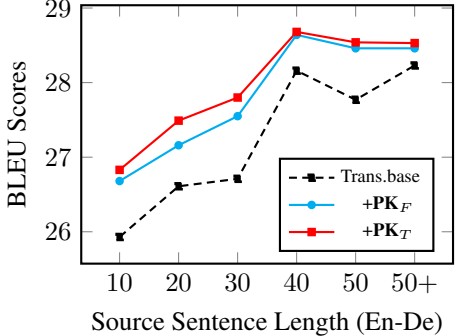
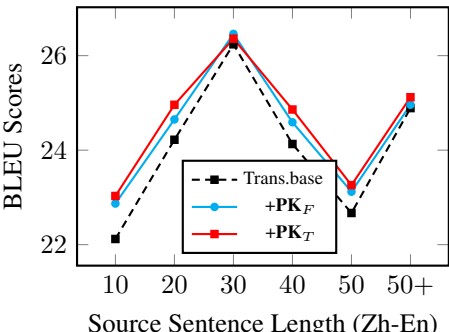

Figure 4: Trends of BLEU scores with different source lengths on the two translation tasks.

## 6 RELATED WORK

There were a variety of methods to combine the advantages of both the dominant NMT and traditional SMT models. Typically, the prior knowledge was used to enhance the NMT, for example, discrete dictionary (Luong et al., 2015; Jean et al., 2015; Arthur et al., 2016; Gu et al., 2016), the limitation of vocabulary (Mi et al., 2016; He et al., 2016; Indurthi et al., 2019), translation of certain terminology (Alkhouli et al., 2018), and translations and interpretable alignments (Garg et al., 2019). Wang et al. (2017a) combined NMT posteriors with SMT word recommendations through linear interpolation implemented by a gating function. Niehues et al. (2016) utilized the SMT model to pre-translate the inputs into target translations and employed the target pre-translations as input sequences in NMT. The combination of NMT and SMT had been also introduced in interactive machine translation to improve the system's suggestion quality (Wuebker et al., 2016). Zhou et al. (2017) presented a neural system combination framework to directly combine NMT and SMT outputs. Different from the direct combination of SMT (Wang et al., 2017a; Wuebker et al., 2016; Niehues et al., 2016; Zhou et al., 2017), the proposed method allows the prior knowledge to be introduced into NMT in a universal neural way. Meanwhile, the prior knowledge representation enables the NMT to semantically generalize the prior knowledge instead of directly combining the probability distribution of the prior translation knowledge.

## 7 CONCLUSION

This article explored a universal representation method to introduce the prior knowledge into the Transformer-based NMT in a universal way. In particular, the proposed method can represent the prior knowledge as a continuous space matrix to semantically generalize instead of directly combining the probability distribution of the prior translation knowledge, thereby enhancing the representation of the input sentence. Experimental results verified the effectiveness of our method on the two wildly used translation tasks. In the future, we will adopt the proposed method to other natural language processing tasks and explore more effective prior knowledge.

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

## A  BILINGUAL TRANSLATION LEXICONS IN SMT

The goal of machine translation is to translate a source sentence $X = x_1^J$ into a target sentence $Y = y_1^I$, where $x_j$ and $y_i$ belong to the source vocabulary $V_x$, and the target vocabulary $V_y$, respectively. In the traditional SMT systems, bilingual lexical translation rules are generally learned directly from the large-scale bilingual parallel data in an unsupervised fashion using the IBM models (Brown et al., 1993; Koehn et al., 2003). These models can be used to estimate the word alignments $a$ and lexical translation probabilities $p_{(a)}(y|x)$ between the words of the two language through the expectation maximization algorithm. First, in the expectation step, the algorithm estimates the expected count $c(x, y)$. In the maximization step, lexical translation probabilities are calculated by dividing the expected count by all possible counts:

$$p_{(l,a)}(y|x) = c(x,y) / \sum_{\hat{y}} c(x, \hat{y}). \tag{13}$$

As a result, there is a list table in which each source word is linked to a set of possible target candidate translations, called bilingual lexicon translation table $\mathcal{T} = \{x, y, p_{(a)}(y|x)\}$ from source language to target language.

