# OpenReview forum: "Prior Knowledge Representation for Self-Attention Networks"
_ICLR.cc/2021/Conference — Reject_

### Official Review · AnonReviewer2 · 2020-10-26
**incremental contributions for NMT and unclear parts**

**Rating:** 3
**Confidence:** 5

**Review:**

Summary:

This submission works on the neural machine translation problem. The authors extend the previous works on leveraging language statistics or prior knowledge (SMT model or whatever) in LSTM based NMT models in self-attention based NMT models, Transformer model. The authors propose two alternatives to incorporate prior knowledge, which are the word frequency information for the monolingual data and the prior translation lexicon information for the bilingual data. These resources are integrated into the hidden representations from the self-attention computations and then the two output hidden representations are gated together for upper computations. The experiments are conducted on two typical NMT datasets: WMT14 En->De and WMT17 Zh->En, the results show that the proposed method can improve the NMT model performances.

General comments:

The submission works on the neural machine translation tasks, which are important in recent years and have achieved great breakouts. This work extends the previous work on how to leverage the information from the statistical data knowledge into the NMT model. Overall speaking, there is no big tech flaws or incorrect points, but this paper is not suitable for ICLR conference, better for NLP related conferences. The main pros and cons of my side are listed as follows.

Strengths:
* The paper writing is good in an overall reading, no big mistakes existed in this submission.
* The experiments are conducted on two different large datasets, with extensive ablation studies to show the effectiveness of each part of the model.

Weaknesses:
* The motivation of this work is not presented in a very clear way. The authors claim that previous works can not explore other prior knowledge in a universal way. This is somehow too overwhelmed, without clear examples or explanations, I feel hard to be convinced of this point. Besides, the authors do not present that the methods utilized in this submission are much better than previous works in terms of the differences and contributions.
* The contribution from both technique insights and the studying problem is limited. First, for the studying problem, how to leverage the information from the statistic data knowledge is widely studied in the MT committee previously and in recent years. The authors must be very familiar with the works conducted on LSTM models about incorporating SMT knowledge or others. Second, as for the technique contribution, the authors propose word frequency information and the translation lexicon knowledge. These two are typical methods used in MT research, which makes the submission not so insightful from the different prior knowledge. Therefore, the contributions of the submission are not important or enough.
* As for the specific utilization of the information, there are also several concerns. The authors seem to emphasize the prior word frequency from the "monolingual" data, which looks unnecessary. In current modeling, there is nothing related to extra monolingual data, but only the source data. The authors want to compare with the lexicon information from the bilingual dataset, but this will mislead the readers to think of the extra monolingual data. As for the bilingual translation lexicon, the extra costs increase and the authors should discuss this.
* There are several unclear parts of the presentation. First, the dimension of the matrix $M$ is not clear. For example, $M_F$ seems to be a $J\times J$, but for a bilingual lexicon, it is $J\times K\times d_{model}$ for $M_T$. Therefore, what is induced $K_F, V_F$ and $K_T, V_T$, and how are they converted to $K$  and $V$. By the way, in the bilingual lexicon, $L$ is leveraged, but not show in $M_T$, this also makes it to be unclear. It supposes that $M$ should be a similar size as $H$, but seems not.
* In current modeling, the prior knowledge is only incorporated in the encoder-side and the training phase, if I understand correctly. This is also limited. I acknowledge that the word frequency information is from the training data, therefore only the training phase is influenced, but the authors should clearly talk and discuss this. By the way, the encoder-side only modeling again makes the contribution to be limited. More efforts are encouraged than this submission.
* One another point is about the experiment results. The authors said that they reimplement the baselines of the Transformer base model and Transformer large model. However, according to the settings they described in the submission, the baselines are reported at a low level, not a very convincing score. The authors used attention dropout and residual dropout as $0.1$, and also they use $8$ GPU cards (since $4096\times 8$ batch size), and checkpoints are averaged. Therefore, according to the experiences, the baseline results should be higher, and it makes me not be convinced about "which makes the evaluation convincing". However, I acknowledge that the method proposed is effective and the results of the proposed method are in reasonable scores, what I want to mention is only the baseline results.

Minor suggestions:
* Section 5.7 about length experiments is not necessary, it is not related to the claim, motivation, contribution of the submission. So it is recommended to remove this part.

In a word, this paper is okay for NLP related conferences, but maybe not enough for ICLR.

---

### Official Review · AnonReviewer4 · 2020-10-26
**Official Blind Review #4**

**Rating:** 5
**Confidence:** 4

**Review:**

This paper proposes a method to introduce **prior knowledge** into Transformer-based sentence encoders, here in the context of neural machine translation (NMT). More concretely, the prior knowledge is represented in the form of a matrix $\boldsymbol{M}$, where each row denotes a vector of prior knowledge associated with each word $x_i$. The prior knowledge matrix $\boldsymbol{M}$ is then represented as a (key, value) pair that can be attended by the query matrix $\boldsymbol{Q}$ (the same query matrix as used in the main NMT component) using a standard Transformer self-attention mechanism. This procedure results in a prior knowledge representation matrix $\boldsymbol{PK}$, which is then combined with the standard Transformer encoder output using a simple gating mechanism.

Here the paper explores two kinds of prior knowledge:
1. a binary mask that zeroes out frequent yet semantically uninformative function words (which here is defined as the top $B$ most frequent words on the training corpus) whilst keeping the content words the same, and
2. a word type-level translation lexicon that defines the **L** most frequent target words for each source token, as obtained from word alignment methods used in statistical machine translation (SMT) systems.

Experiments on WMT English-German and Chinese-English translation datasets suggests that the prior knowledge-augmented model outperforms the standard Transformer models without any prior knowledge representation, and compares favourably to other alternative methods of injecting prior knowledge as proposed in prior work.

**Pros:**
1. The question of how we can inject prior knowledge into neural sequence-to-sequence models is an important and interesting research question. Rather than learning everything from scratch, injecting prior knowledge may lead to more data-efficient models, and can also encourage better adequacy (since neural seq2seq models often produce fluent yet inadequate generations).
2. The proposed approach is empirically validated on two machine translation benchmarks, and obtains improvements in both datasets. Furthermore, the improvements are consistent across both base and large Transformer setups.

**Cons:**
1. Despite the positive points above, my main concern with this paper is regarding the clarity. In particular, the introduction does not clearly explain **where exactly** previous work on injecting prior knowledge falls short. For instance, the last paragraph of page 1 mentions "..these unique methods make it difficult to explore other prior knowledge in a universal way and to determine which of the prior knowledge and the unique method this improvement comes from". This sentence is quite hard to understand---the paper does not really explain how the proposed approach is more universal than previous ones. In particular, the paper only explores two kinds of prior knowledge, and only for the task of machine translation---any extensions to other prior knowledge seem to need careful manual design. Furthermore, the beginning of page 2 mentions that "Meanwhile, these [previous] studies directly utilized the probability distribution of the prior knowledge and lacked the neural network's ability to semantically generalize ...". I fail to see what exactly the proposed approach has to do with semantic generalisation. Overall, the introduction could benefit from a substantial reframing to better motivate why we need better ways of injecting prior knowledge. Furthermore, the paper can also benefit from *substantial copy-editing*, as there are many grammatical errors that make the paper a bit hard to read.

2. The analysis section does not seem particularly insightful. In particular, the probing experiment is limited to predicting bag-of-words (which seems like a very artificial task), while the analysis on the choice of hyper-parameters (Section 5.6) does not strike me as particularly interesting (for instance, in Figure 3, *any* hyper-parameters above 0 do better than 0, and the differences between different hyper-parameters are fairly minimal). The fact that NMT systems do worse on longer sentences has also been established before in prior work, and is thus not particularly insightful. I have some suggestions for analysis as detailed in the **Suggestions** section below.

3. The paper has some errors in terms of notational inconsistencies and grammatical mistakes that make it a bit hard to read. For instance, the prior knowledge hyper-parameter for bilingual translation lexicon is denoted as $L$ in page 3 (between equations 8 and 9), but the same hyper-parameter is denoted as $K$ in the caption of Figure 3 (page 7). Some of the grammatical errors are detailed below.

**Suggestions:**
Rather than the current set of analysis experiments, I have some suggestions for further analysis that may be more informative:
1. It would be interesting to analyse the translation performance on **rare words**. The prior knowledge in terms of word translation lexicon may help translate rare words better, since the standard NMT systems tend to hallucinate fluent yet inadequate outputs.
2. It would also be interesting to see the effects of injecting prior knowledge on **adequacy**. Does the prior-injected model suffer from less hallucination? Or does it more faithfully translate all the content words in the source sentence (as opposed to skipping and not translating some of the content words)?
3. It would also be interesting to qualitatively show some examples where the model outputs differ between the baseline and the proposed approach.

**Questions:**
1. The frequency prior assumes that the top $B$ most frequent words correspond to function words. Did you verify whether this is really the case?
2. What happens if you inject both prior knowledge into the model, rather than just choosing one?

**Typos / Grammatical Mistakes / Minor Clarification:**
1. Abstract: "... it gradually *learning* ..." -> should be "it gradually learns".
2. The first paragraph in page 1 mentions "... in many natural language processing tasks pairs". Not sure what is meant by tasks pairs here.
3. Page 1: "... explored the prior knowledge which has the *stringer* ability" -> should be "stronger".
4. Page 1: "... the phrase translation rules *was*" -> should be "were".
5. Page 1: "... translations of *infrequency* words" -> should be "infrequent".
6. The last paragraph of page 2 mentions "Also, $\mathbf{M}$ is packed into a key and value ...". What exactly is meant by "packed into a key and value matrix pair"?
7. Page 4: "where the sigmoid is an *active* function" -> "activation".

---

### Official Review · AnonReviewer1 · 2020-10-28
**Details and effectiveness of the proposed approach are not very clear**

**Rating:** 4
**Confidence:** 4

**Review:**

This paper presents a method for introducing prior knowledge into Transformer models. More specifically, the authors propose to use an additional self-attention block to incorporate prior knowledge about the word frequency and translation lexicon and use a gating mechanism to combine its output with that of the standard sefl-attention block. Experiments are conducted using English-to-German and Chinese-to-English translation datasets, and the results show the effectiveness of the proposed approach.

The details of the proposed approach are not very clear. The authors state that matrix M is packed into a key and value matrix pair {K, V}, but it is not really clear how exactly these two matrices are produced from M. I think they should describe this process more concretely in Sections 3.1 and 3.2.

I also think that the authors should test simpler baselines as well. It seems to me that such prior knowledge can be incorporated into the standard Transformer model in the same manner as positional encoding or expanded input embeddings. What is the advantage of the proposed approach over these baseline methods?

In Equation (5), $B$ is treated as a set, but it is treated as an integer in the previous paragraph.

The manuscript contains many grammatical errors and is not ready for publication. I suggest that the authors proofread their manuscript much more carefully before submission. Here are some of the errors in the first page:
 - gradually learning -> gradually learns?
 - on WMT14 -> on the WMT14?
 - Gehring et al. (2017) -> (Gehring et al., 2017)?
 - then added to -> then added to the?
 - mistraslate -> mistraslates?
 - infrequency -> infrequent?
 - were employed -> was employed?
 - rules was -> rules were used?

---

### Author Response · Authors · 2020-11-24
**General Response**

Thanks for your valuable and insightful suggestions! As you said, there are indeed many places to be improved:
1) Unclear how exactly these two matrices are produced from M;
2) The advantage of the proposed method seems to be unclear compare to the existing works;
3) The motivation needs to be further refined and more sufficient argument;
4) Some targeted experiments  (i.e., rare words, adequacy, and case study) will be interesting;
5) Exploring the orthogonality of multiple prior knowledge;
6) Needing more experimental analysis and discussion for the proposed method.
7) Others ... ...

Also, we are very sorry that the current writing has brought unnecessary review burden to Reviewers
In short, we will further improve our work accordingly.
Thanks a lot again!

---

### Decision · Program_Chairs · 2021-01-07
**Final Decision**

**Decision:**

Reject

**Comment:**

This paper proposes to incorporate additional prior knowledge into transformer architectures for machine translation tasks. The definition of problem is reasonable,  despite the fact that there is a long thread of work on adding knowledge of different types into neural architectures of NMT. The proposed model, however, needs to be better motivated, as to why the same thing cannot be done in a simpler way in the framework of transformers.  Judging from the exposition and the experiments, the proposed model is neither novel or empirically significant enough. The writing needs to be greatly improved to get rid of the grammatical errors and notational inconsistency.

I’d suggest to reject this paper